# Modelling Neural Disorders with the *D. melanogaster* Larval Peripheral and Adult Dopaminergic Systems

**DOI:** 10.3390/biom15121677

**Published:** 2025-12-01

**Authors:** Daniel Tendero-Lopez, Maria Dominguez, Mario Aguilar-Aragon

**Affiliations:** Instituto de Neurociencias, Consejo Superior de Investigaciones Científicas, Universidad Miguel Hernández (CSIC-UMH), 03550 Sant Joan d’Alacant, Spain; dtendero@umh.es

**Keywords:** neurological disorders, *Drosophila melanogaster*, peripheral nervous system, dopamine

## Abstract

The increasing prevalence of neurological disorders highlights the need for disease animal models to elucidate the underlying biomolecular and cellular mechanisms of disease and to facilitate studies aimed at developing effective treatments. The fruit fly *Drosophila melanogaster*, at both larval and adult stages, can serve as an effective model for different human-relevant neurological diseases. Larvae are particularly suited for studying peripheral nervous system disorders, such as Charcot–Marie–Tooth and amyotrophic lateral sclerosis, while adults enable investigations of higher-order cognitive functions and age-related conditions, including Parkinson’s disease and depression-like behaviours. Combining larval and adult models offers a complementary framework to dissect the biomolecular pathways of neurological disorders and accelerate preclinical research.

## 1. Introduction

Neurological disorders constitute a growing global health challenge. Currently, over 1 in 3 people are affected by neurological conditions, representing the leading cause of illness and disability worldwide [1]. These conditions can arise in, or affect, both the central nervous system (CNS) and peripheral nervous system (PNS), often involving complex and interconnected pathophysiological mechanisms.

Given the growing burden of neurological disorders, there is an urgent need for accessible and genetically tractable animal model systems to elucidate the underlying disease mechanisms and to enable preclinical studies aimed at developing effective treatments [2].

The juvenile (larvae) and adult stages of the fruit fly *Drosophila melanogaster* have emerged as powerful and complementary experimental systems for dissecting the fundamental mechanisms of human-relevant neurological diseases [3,4,5]. Each developmental stage offers unique advantages and constraints that can be strategically leveraged depending on the specific disease under investigation.

The larval stage is particularly well-suited for modelling neurological diseases involving the PNS. The reasons include a relatively simple nervous system [6], the accessibility of peripheral nerves for high-resolution imaging and electrophysiological studies [6,7,8,9], and the repertoire of simple and robust behaviours [10,11,12,13]. Larvae are also valuable for the preclinical preselection and testing of therapeutic strategies [14,15]. Their sensory neurons, particularly the multidendritic (md) Class 4 neurons (C4da), are valuable for research because they are large, highly branched, and easy to genetically manipulate and image [16].

The adult stage provides significant opportunities for modelling complex neurological diseases involving high-order cognitive functions, such as memory and learning [17,18], as well as complex behaviours [19] and studies related to ageing [20]. The dopaminergic system plays crucial roles in promoting survival, health, and rewards, as well as contributing to memory consolidation, learning, motion, and age-related neural changes. By effectively modelling dysfunction in this system, one can gain valuable insights into dopamine-related neurological disorders [21].

This narrative review examines the complementary use of *D. melanogaster* across its life stages to model neurological diseases, highlighting larval-stage studies of peripheral neuropathies and adult-stage investigations of dopaminergic-related disorders. *Drosophila* models have also been widely applied to study polyglutamine expansion disorders, such as Huntington’s disease and spinocerebellar ataxias, which share conserved pathogenic mechanisms with mammalian systems [22,23]. Although these diseases are not covered in the present review, they further exemplify the fly’s versatility as a model for neurodegeneration.

Relevant articles were identified through searches in PubMed and Google Scholar, using general keywords such as “*Drosophila* larvae and adult models”, “neural disorders”, “dopamine-related disorders”, and more specific terms including “Charcot–Marie–Tooth”, “diabetic and chemotherapy-induced neuropathies”, “amyotrophic lateral sclerosis”, “depression-like disorder”, “Parkinson’s disease”, and “attention-deficit hyperactivity disorder (ADHD)”. The search covered the period from 2010 to 2025. Together, these approaches emphasise the fruit fly’s versatility and its capacity to elucidate the molecular mechanisms underlying a broad range of nervous system pathologies.

## 2. Modelling Peripheral Neuropathies in the Larval Peripheral Nervous System of *D. melanogaster*

The *D. melanogaster* larval peripheral nervous system provides a tractable experimental model that allows for experimental consistency and comparability. The larval peripheral nervous system is a highly stereotyped network [24], crucial for sensory perception and motor coordination.

The head contains chemosensory organs: pharyngeal sensilla, the cool-sensing terminal organ, and the Bolwig organ for photoreception and circadian regulation [25,26]. Along the body wall, repetitive neuronal sensory clusters in each hemisegment of the body contain type I mechanosensory neurons—external sensory (es) and chordotonal (cho) organs—and type II md neurons [24]. The md neurons are characterised by their multiple dendritic projections, and include bipolar dendrite (bd), tracheal dendrite (td), and dendritic arborization (da) subtypes [24]. Da neurons are further divided into classes 1–4, ranging from proprioceptors and gentle-touch sensors to multimodal nociceptors detecting thermal, mechanical, chemical, and UV light exposure [27,28,29]. Among the four classes, the nociceptive C4da neurons (C4da) stand out as one of the most utilised neuronal subtypes for studying neurological diseases in larvae.

### 2.1. Charcot–Marie–Tooth

Charcot–Marie–Tooth (CMT) is a group of inherited motor and sensory neuropathies, with a prevalence of 17.69 per 100,000 [30]. The clinical phenotype is defined by demyelination, length-dependent axonal degeneration, or a combination of both, affecting peripheral motor and sensory nerves [31,32]. These pathological changes manifest during childhood or adolescence, generating muscle weakness in the extremities, foot and hand deformities, reduced or absent tendon reflexes, and varying degrees of sensory loss [32].

A summary of the *D. melanogaster* models of CMT discussed in this section is presented in Table 1. One major pathogenic mechanism involves mutations in aminoacyl-tRNA synthetases, particularly the glycyl–tRNA synthetase (*GARS*). In CMT *D. melanogaster* larval models, a mutant form of human GARS disrupts neuromuscular junction integrity, causes progressive muscle denervation, and impairs neuronal function via translational dysregulation [33,34]. Expression of mutant *GARS* in C4da neurons reduces dendritic coverage without affecting cell bodies and decreases protein translation, lowering newly synthesised proteins by ~30–50% [34] (Figure 1A).

Further elucidating this mechanism, Zuko et al. (2021) [33] demonstrated that *GARS1* mutations disrupt translation elongation by depleting available Glycine tRNA, causing ribosomes to pause at glycine codons and triggering activation of the integrated stress response, shutting down global protein synthesis (Figure 1A). Importantly, overexpression of Glycine tRNA partially rescued the observed phenotypes in the GARS model, including translational defects, larval muscle denervation, larval sensory neuron morphology defects, adult motor deficits, developmental lethality, and shortened lifespan (Figure 1A).

CMT can also be caused by missense mutations in the *RAB7A* gene, which encodes a small GTP-ase involved in intracellular trafficking, particularly in the endo-lysosomal system [35]. RAB7 is a highly conserved protein; the human form shares 76% identity and 95% similarity with its single *D. melanogaster* ortholog. When the human mutant *RAB7* form is expressed in sensory neurons, larvae exhibit reduced temperature perception, as well as a decreased nociceptive response compared to controls. These phenotypes recapitulate aspects of the sensory deficits observed in human CMT patients, suggesting conservation in the pathogenic mechanisms [36]. Despite the presence of sensory defects, the dendritic morphology of affected neurons remains unaltered in class I sensory neurons, as demonstrated by Janssens et al. (2014) [36]. This finding indicates that the observed functional impairments are not due to structural abnormalities. Instead, axons from mutant neurons showed abnormal accumulation of RAB7-positive vesicles with reduced stationary time, suggesting that trafficking defects might underlie the sensory impairments (Figure 1B).

Studies in *D. melanogaster* models of CMT disease demonstrate that mutations in *GARS* disrupt global protein synthesis and induce neuromuscular degeneration, paralleling translational defects observed in human and mouse models [37]. Similarly, pathogenic *RAB7* mutations impair endolysosomal trafficking in flies, mirroring phenotypes and transport deficits seen in human CMT patients [38]. These findings have provided critical insights into how disruptions in protein synthesis and intracellular trafficking contribute to neurodegeneration.

### 2.2. Peripheral Diabetic Neuropathy and Chemotherapy-Induced Neuropathy

Diabetes is currently the largest global epidemic, with 537 million adults affected in 2021, projected to rise to 643 million by 2030 and 783 million by 2045 [39]. Both type I and type II diabetes can lead to secondary complications, with diabetic peripheral neuropathy (DPN) being the most prevalent. DPN involves nerve damage, primarily in the lower limbs, and often leads to nociceptive sensitization, resulting in sensory loss, pain, foot ulcers, and lower limb amputations, contributing to significant morbidity [40,41]. Similarly, chemotherapy-induced peripheral neuropathy (CIPN) also damages limb nerves, causing allodynia, hyperalgesia, and other neuropathic symptoms [42].

Several *D. melanogaster* larval models have been developed to investigate mechanisms underlying DNP and chemotherapy-induced peripheral neuropathy (summarised in Table 1). Im et al. (2018) [43] examined the role of impaired insulin signalling in nociceptive dysfunction by analysing mutants of the insulin-like receptor (InR). *InR* mutant larvae display persistent thermal hyperalgesia following UV-induced injury, suggesting a failure to resolve acute nociceptive sensitization despite normal baseline sensitivity, resembling early DPN. Using both type 1 and type 2 diabetes models—silencing insulin-producing cells and a high-sugar diet, respectively—both models also displayed persistent thermal hyperalgesia. In the type 1 model, UV injury reduced dendritic length and branch number in C4da neurons, whereas the type 2 model showed no significant morphological changes (Figure 2, top panel).

To assess insulin signalling in sensory neurons, *InR* knockdown in md neurons reproduced the diabetic-like persistent hyperalgesia phenotype, with reduced baseline dendritic length and unchanged dendritic length post-injury despite increased branching [43]. This persistent thermal hyperalgesia correlated with elevated calcium responses in C4da neurons. Constitutive *InR* activation caused acute-phase hyposensitivity, whereas restoring insulin-like signalling specifically in md neurons rescued persistent nociceptive hypersensitivity [43].

In addition to thermal nociceptive sensitization, InR in peripheral md neurons also regulates diabetic mechanical nociceptive hypersensitivity, but only in the type II diabetic model [44,45].

Peripheral neuropathies that include nociceptive sensitization could also be induced by anti-cancer drugs. Some of them are paclitaxel and vincristine, drugs that bind along the microtubule lattice and suppress microtubule dynamics, leading to cell cycle arrest and tumour cell death [46]. A *D. melanogaster* larvae model of CIPN generated by feeding larvae with either paclitaxel or vincristine was reported to cause axonal injury and loss, independent of apoptosis [47,48,49] (Figure 2, bottom panel).

CIPN models enable the study of the mechanisms and the exploration of potential preclinical treatments to relieve symptoms. In 2020, Kim et al. demonstrated that ectopic expression of the PTEN-induced kinase 1 (*Pink1*) in C4da peripheral neurons alleviates paclitaxel-induced thermal hypersensitivity in *D. melanogaster* larvae by restoring mitochondrial homeostasis (Figure 2, bottom panel). In addition, *Pink1* silencing worsens sensitivity, supporting its protective role [50]. Subsequent studies have shown that the Pink1 activator niclosamide mitigates paclitaxel-induced mitochondrial dysfunction in a Pink1-dependent manner, without reversing C4da neuron arborization defects, and similar effects were observed in human SH-SY5Y cells [51]. In a vincristine-induced neuropathy model, thermal hypersensitivity and C4da dendritic structural alterations were also observed, and silencing the mitochondrial pyruvate dehydrogenase complex (PDH) rescued the mitochondrial and sensory abnormalities caused by vincristine [49] (Figure 2, bottom panel). These models have provided valuable insights for identifying potential therapeutic targets for CIPN.

More recently, Im et al. (2024) [52] reported that PDE701, a diphenyl ether derivative from a marine sponge, could act as a therapy against CIPN. This compound induces mitophagy, a specific type of autophagy that selectively degrades mitochondria, ensuring proper mitochondrial quality control and turnover (Figure 2, bottom panel). The compound alleviates mitochondrial dysfunction in both in vitro human SH-SY5Y cells and in vivo in *D. melanogaster* larvae and reduces chemotherapy-induced thermal hyperalgesia.

Beyond mitochondrial regulators, additional protective mechanisms have been identified in response to paclitaxel exposure [47,48,53]. That is the case of nicotinamide mononucleotide adenylyltransferase (Nmnat), which can block the axonal degeneration induced by the drug [47] (Figure 2, bottom panel). Nmnat is required for maintenance of sensory neuron integrity and for maintenance of nociceptive integrity, and its overexpression in C4da neurons mitigates the nociceptive hypersensitivity of the model (Brazil et al., 2018 [48]). Integrins represent another protective mechanism that contributes to sensory neuron resilience against paclitaxel-induced neuropathy [53]. Paclitaxel disrupts integrin-mediated adhesion to the extracellular matrix, leading to defects in dendritic self-avoidance characterised by wide branch crossing. The co-overexpression of αPS1 and βPS integrins in nociceptive neurons reduces these defects, indicating that integrins help maintain a dendritic structure, protect against paclitaxel-induced damage, and prevent thermal nociceptive sensitivity (Figure 2, bottom panel). This protective effect of integrin overexpression is linked to the fact that paclitaxel impairs the integrin trafficking in larvae by disrupting the endosomal–lysosomal pathways, which precedes the early morphological signs of degeneration [53].

*D. melanogaster* models of diabetic and chemotherapy-induced peripheral neuropathy recapitulate key features observed in mammals, including persistent nociceptive sensitization, dendritic and axon degeneration, mitochondrial dysfunction, and deficits in sensory integration [54,55]. Interventions that modulate mitochondrial quality control, axonal degeneration, hyperalgesia, or integrin function in fruit flies offer key experimental validation for therapeutic approaches that may extend to human peripheral neuropathies, highlighting the translational importance of these conserved mechanisms.

### 2.3. Amyotrophic Lateral Sclerosis

Amyotrophic lateral sclerosis (ALS) is the most common neurodegenerative disease affecting motor neurons. It is characterised by progressive muscle weakness and atrophy, with a median survival of 2 to 4 years after diagnosis. It is a relatively rare disease, with an incidence that ranges from 0.73 to 2.25 per 100,000 person-years [56].

Although ALS has been considered a specific motor neuron disease, in recent years, the paradigm is changing to consider it a multisystem disorder, including non-motor symptoms like dysautonomia, cognitive impairments, and sensory problems, suggesting that the sensory system may contribute to the pathogenesis of the disease [57,58]. The *D. melanogaster* larval peripheral nervous system has emerged as a powerful model to study ALS-associated molecular mechanisms to discover novel biomolecular pathways. The main *D. melanogaster larval* ALS models are listed in Table 1.

A hallmark of ALS pathology is the mislocalization and aggregation of TAR DNA-binding protein 43 (TDP-43), a conserved RNA-binding protein involved in RNA metabolism and transport [59]. In larvae, the ALS model based on the overexpression of TAR DNA-binding protein-43 homologue (*TBPH*), the ortholog for human *TDP-43*, leads to excessive dendritic branching in sensory neurons C4da, while the downregulation reduces branching, demonstrating its critical role in neuronal morphogenesis via cell-autonomous mechanisms [60]. Under physiological conditions, TBPH is predominantly localised in the cytoplasm during larval stages, with only a small fraction in the nucleus, but TBPH changes its localization from cytoplasm to the nucleus at the pupal stage [61]. This subcellular redistribution is regulated by cytosolic calcium levels, as increased calcium in pupae correlates with reduced cytoplasmic/nuclear TBPH ratios, whereas mutations that reduce intracellular calcium led to increased TBPH cytoplasmic accumulation [61].

Interestingly, inhibition of Valosin-Containing Protein (VPN), a ubiquitin-dependent ATPase whose human homologue is linked to ALS, causes TDP-43 to relocalise to the nucleus without altering total protein levels [62].

Notably, the ALS-linked mutant *TDP-43 G287S* also relocalises from the cytoplasm to the nucleus during development, but the response to calcium fluctuations is reduced, suggesting impaired calcium-dependent transport mechanisms. Several calcium-dependent regulators, including Calmodulin, Protein kinase C, and Calpain-A (CalpA), are essential for proper TBPH nuclear localization [61]. RNAi silencing of these regulators leads to TBPH cytoplasmic retention in pupal sensory neurons. In addition, core nuclear transport proteins, including Importin α1, α3, β1, Importin 7, and Transportin-SR, are required for efficient nuclear import of TBPH. Overexpression of Importin α3 and β1 enhances nuclear translocation, suggesting that these nuclear transport biomolecules modulate TBPH localization dynamics [61].

CalpA emerges as a critical mediator linking calcium signalling to the nuclear import machinery. Co-overexpression experiments demonstrated that CalpA enhanced nuclear accumulation of Importin α3, whereas *CalpA* knockdown blocked the calcium-induced nuclear localization of Importin α3, and as a result, TDP-43 (TBPH) also failed to enter the nucleus [61]. These findings reveal a biomolecular regulatory axis involving cytosolic calcium, CalpA, and importins that control the subcellular distribution of TDP-43.

Another *D. melanogaster* larval model of ALS is based on the GGGGCC RNA microsatellite repeat expansion in the *C9orf72* gene, a major genetic cause of the disease [63,64]. Expressing UAS-(GGGGCC)48 RNA in C4da neurons leads to normal dendrites at early third instar larvae but severe defects in the late third instar stage, including 42% fewer distal intersections and 53% loss of higher-order branches. These neurons form complex arbours initially, but later fail to grow and degenerate [65].

The data suggest that the mechanism of branching defects could be due to altered RNA transport granule function. Downregulation of *dFMR1*, a component of mRNA transport granules or *Orb2*, a transport granule protein involved in the local translation of neuritic RNAs, rescues the dendritic branching defects generated by the expanded microsatellite repeat in *D. melanogaster* sensory neurons. In contrast, the upregulation of either *dFMR1* or *Orb2* generates an increase in the branching abnormalities, indicating that modulation of transport granule molecules impacts the branching defects [65].

Repeated sequences such as GGGCC can be transcribed in both sense and antisense directions, producing repeat RNAs that frequently form nuclear concentrations where various RNA-binding proteins are trapped. Some of these repeated RNAs can exit the nucleus and undergo repeat-associated non-ATG (RAN) translation, generating distinct dipeptide repeat proteins (DPRs) [66]. In 2020, Park and collaborators showed that arginine-rich DPRs, specifically PR36 and GR36, induce loss of dendritic branch points in *D. melanogaster* C4da da neurons [67]. These arginine-rich DPRs also disrupt plasma membrane supply and reduce the number of Golgi outposts (GOPs), which are structures essential for maintaining dendritic architecture. The GOPs could be regulated by secretory pathway-related genes, with the Cyclic-AMP response element binding protein A (*CrebA*) being one of the most important. PR36 localises to the nucleus and downregulates the expression of *CrebA*, while GR36 does not change *CrebA* mRNA levels, indicating a possible different mechanism of toxicity [67]. Overexpression of *CrebA* restores GOPs and plasma membrane supply in neurons that express *PR36* but fails to recover dendritic morphology or prevent degeneration, suggesting that toxicity induced by PR36 involves both *CrebA*-dependent and independent pathways [67].

ALS could also be caused by mutations of the copper-zinc superoxide dismutase 1 (*SOD1*) gene. In the larval Sod1-ALS model [68], motor impairments occur in both early and late larval stages. While late-stage larvae exhibit clear motor neuron degeneration, early-stage larvae show only minor changes in motor neuron structure and function [69]. Analysis of the intact motor circuit in the early larval stage revealed a defect in sensory feedback before motor degeneration that could cause the altered motor activity. Remarkably, cell-autonomous activation of BMP signalling in proprioceptive sensory neurons, which are critical for relaying muscle contractile status to the central nerve cord, fully rescues early-stage motor defects and partially rescues late-stage motor function to extend lifespan, highlighting the critical role of nonmotor neurons in ALS progression [69].

These *D. melanogaster* ALS models, including those based on TBPH mislocalization, *C9orf72* repeat expansions, and *Sod1* mutations, replicate key pathological features such as dendritic and synaptic defects, dysfunction of nucleocytoplasmic transport, impairment of axonal transport of TDP-43 mRNA granules, and motor and sensory neuron dysfunctions observed in human patients, mouse models, and human stem cell models [70,71,72]. The molecular mechanisms uncovered in flies are conserved in ALS patients, providing vital insights into disease pathogenesis. These models have also revealed candidate therapeutic targets, including importins, *FMR1*, and the human orthologs of *CrebA*, that have potential translational implications for intervening early in ALS progression in humans.

## 3. Modelling CNS Neuropathies Using the Dopaminergic System of *D. melanogaster*

The adult *D. malanogaster* brain contains approximately 127 dopaminergic neurons distributed across eight clusters in each hemisphere. Each cluster consists of 4 to 13 individual neurons. Additionally, single dopaminergic neurons can be found in up to four distinct regions: PPD, PPL3, PPL4, and PPL5 [73].

Despite their relatively small quantities, dopaminergic neurons have extensive projections throughout the brain, exerting broad functional influence. Dopamine plays a crucial role in regulating behaviours, including basal locomotion, sleep, arousal, light perception, circadian rhythms, courtship, feeding, learning, aversive conditioning, aggression, and social spacing. Furthermore, dopamine is essential for higher-order cognitive processes such as memory, addiction, attention, decision-making, and appetite regulation [74].

### 3.1. Depression-like Disorder

Depressive disorder is a prevalent mental health condition worldwide, characterised by symptoms such as persistent sadness, feelings of emptiness, diminished interest or pleasure in daily activities, impaired concentration, hopelessness, disrupted sleep and appetite, reduced energy, and, in severe cases, suicidal thoughts. Globally, it is estimated that approximately 5% of adults experience depression, with women being more affected compared to men, according to the World Health Organization. The pathophysiology of depression disorder includes the dysregulation of neurotransmitters, including serotonin, norepinephrine, and dopamine [75,76,77].

*D. melanogaster* has been employed to study induced depressive-like behaviours and various depressive-like symptoms. This review focuses on the *D. melanogaster* models that investigate the role of dopamine in mediating depression-associated phenotypes, summarised in Table 2. Depression-like phenotypes could be induced by drugs, such as levodopa (L-DOPA), the metabolic precursor of dopamine, and Chlorpromazine (CPZ), an antipsychotic drug [78,79,80]. Treatment with either L-DOPA or CPZ in flies induces a reduction in appetite, as well as a decrease in sexual activity, with the males being more affected by the drugs [80]. Flies treated with L-DOPA also display negative geotaxis deficits, as assessed by the climbing assay paradigm [79] (Figure 3). Transcriptomic analysis has revealed that male flies treated with CPZ exhibited distinct gene expression changes compared to controls. Among the Differentially Expressed Genes (DEGs), the largest group identified based on Gene Ontology analysis comprised genes involved in metabolic processes, followed by genes involved in single-organism processes [80]. Notably, *CG4269*, a socially responsive gene whose molecular function is still unknown, was strongly upregulated in CPZ-treated males. In contrast, *CG6821*, which encodes Larval serum protein 1 gamma (Lsp1γ), a component of the tyrosine metabolism pathway, was markedly downregulated in depressive-like males following CPZ treatment. Interestingly, L-DOPA-treated males also exhibited overexpression of *CG4269*, whereas in females, *CG6821* was the most strongly downregulated gene compared to untreated controls. Unlike the CPZ condition, however, *CG6821* expression remained unchanged in L-DOPA-treated males, highlighting a treatment and sex-specific difference in gene regulation [79]. Ahn et al. (2021) [78] investigated the potential antidepressant effects of *C. sativa* (hemp) seed ethanol extract (HE) using the CPZ-induced depression model in adult flies. CPZ-induced depression was associated with decreased movement during the subjective daytime and a disrupted circadian rhythm measured using the *Drosophila* Activity Monitor (DAM) system. Depressive-like flies also exhibited a significant reduction in different locomotor behaviours, including distance movement, velocity, and mobility, compared to untreated controls. Remarkably, the combination treatment of CPZ and HE significantly enhanced the subjective daytime activity and improved the locomotor behaviour (Figure 3).

At the molecular level, CPZ-induced depression-like behaviour reduced Dop1R1 and 5-HT1A expression. In addition, co-treatment with 1.5% HE significantly upregulated both receptors, highlighting HE’s potential to counteract CPZ-induced dopaminergic and serotonergic dysregulation [78]. A deeper analysis of depression-related neurotransmitters showed that dopamine and L-DOPA levels were significantly lower in CPZ-treated flies compared to untreated flies. In contrast, HE treatment in CPZ-depressive flies produced a dose-dependent increase in both L-DOPA and dopamine, with the highest HE dose significantly elevating L-DOPA compared to controls [78].

Depression-like models in *D. melanogaster* can also be established through chronic unpredictable mild stress (CUMS), a paradigm in which animals are exposed to a variety of unpredictable stressors. This strategy mirrors the etiopathogenesis of depression. In these models of depression-like behaviour, adult flies are exposed to different stressors during 10 days, including heat, cold, sleep deprivation, and starvation [81,82] (Figure 3).

Following the exposure to these stressors, flies that suffered the CUMS displayed several depression-like phenotypes, including shorter swimming times and increased immobility in the free-swimming test (FST), heightened aggression, and deficits in mating behaviours, such as longer latency to copulation, reduced copulation duration, and lower male fertility. Additional features include decreased sucrose preference, reduced body weight, more time in the dark compartment in the light–dark box test, and diminished levels of serotonin and dopamine. Importantly, dopamine reduction negatively correlates with aggression and immobility, while positively correlating with mating time, sucrose preference, and light–dark box performance [81].

Following these findings, researchers treated CUMS flies subjected to fluoxetine (FLX), a selective serotonin reuptake inhibitor, to confirm the antidepressant-like effects of the model [81]. In another study, γ-oryzanol (ORY) was administered under the same conditions to explore its potential antidepressant effect [82]. Both drugs improved different depression-like phenotypes induced by CUMS, attenuating aggression, preventing impairment of mating performance, and reducing anxiety (Figure 3). Interestingly, FLX and ORY partially restored dopamine levels and completely restored serotonin levels, confirming that the recovery was caused by the antidepressant effects of the drugs and not by random side effects [81,82].

Chronic stress is known to impair learning and memory and increase susceptibility to psychiatric disorders such as depression, although the underlying mechanisms remain elusive. In *D. melanogaster*, chronic-stress-induced learning deficits (CSLD) are persistent and accompanied by other depression-like behaviours. Using a paradigm of repeated mechanical stress for generating chronic stress treatment (CST), flies give rise to impaired olfactory learning and middle-term memory, as well as behavioural impairments, without causing body or brain damage, indicating selective disruption of associative processes [83] (Figure 3). Crucially, dopamine (DA) signalling was identified as a central mediator.

Reduction in DA using 3-iodotyrosine (3-IY) during the final phase of CST alleviated learning deficits (Figure 3), whereas elevating DA with L-DOPA plus carbidopa worsened impairment, suggesting that excess DA under stress is maladaptive [83]. At the receptor level, partial downregulation of *Dop1R1* (*Dop1R1dumb2/+* heterozygotes) did not alter baseline learning but protected against CSLD. Conversely, overexpression of *Dop1R1* in mushroom bodies (MBs) induced learning deficits even in the absence of stress, highlighting the importance of receptor dosage. Finally, neuronal manipulations revealed that dopaminergic activity is both necessary and sufficient for CSLD. Blocking DA neuron transmission during CST prevented deficits, while their chronic activation with TrpA1 alone was sufficient to induce them [83].

Further building on these findings, it was demonstrated that chronic stress not only involves DA-mediated modulation but also impairs neuronal autophagy [84]. Excessive stress-induced dopamine activity disrupts autophagic flux, leading to learning deficits, whereas neuropeptide F (NPF) signalling to dopamine neurons preserves autophagy and confers resilience [84] (Figure 3). These results suggest that maladaptive DA signalling under stress compromises both synaptic and cellular homeostasis, while protective NPF input stabilises dopamine neuron function, highlighting a critical balance between stress vulnerability and resilience mechanisms.

These *D. melanogaster* models of depression, based on pharmacological induction, chronic stress, and dopaminergic dysregulation, replicate key features of human depressive disorder, including altered motivation, disrupted circadian rhythms, impaired cognition, and neurotransmitter imbalances. The conservation of dopamine signalling validates these models for dissecting the molecular basis of depression and testing potential antidepressants. For example, studies show that cannabis oil improves depressive-like behaviours in chronic complex stress mice [85], and FLX reversed depressive-like phenotypes in poststroke depression mice [86], highlighting flies as efficient tools to complement mammalian research in antidepressant therapies.

### 3.2. Attention Deficit Hyperactivity Disorder

Attention-Deficit Hyperactivity Disorder (ADHD) is a neurodevelopmental disorder characterised by inattention, hyperactivity, and impulsive behaviour [87]. Its global prevalence is estimated to be 5–7% worldwide [88]. ADHD has a strong genetic basis, with heritability estimated at approximately 80% [89]. However, environmental risk factors also play a role in its development, including maternal pre-pregnancy obesity and smoking during pregnancy [90].

A summary of the *D. melanogaster* models developed to study ADHD-related mechanisms is presented in Table 2. Among the genetic factors, several risk loci have been identified that affect neurotransmitter regulation, particularly through alterations in dopamine signalling at synapses, thereby linking dopaminergic dysregulation to ADHD pathophysiology [91]. Related to this alteration in dopamine signalling, in 2015, a study was performed to explore a dopamine-associated behavioural endophenotype in adult fruit flies [92]. Pan-neuronal knockdown of three ADHD-related gene orthologs, Dopamine transporter (*DAT*), Latrophilin (*LPHN3*), and Neurofibromin (*Nf1*), produced elevated activity and reduced sleep during darkness. Notably, these phenotypes occurred without changes in dopaminergic neuron counts, suggesting altered dopamine function rather than cell loss. Importantly, feeding flies with the ADHD stimulating medication methylphenidate (MPH) reversed these behavioural abnormalities, further solidifying their link to dopamine signalling [92].

A subsequent study tested whether genes implicated in intellectual disability are associated with ADHD using large Genome-Wide Association Study (GWAS) datasets. Gene-level analyses highlighted *MEF2C*, *TRAPPC9*, and *ST3GAL3* as key contributors [93]. To examine their functional roles, the authors knocked down *Mef2* and *TRAPPC9* in *D. melanogaster* neurons, including pan-neuronal, dopaminergic, and circadian populations. Pan-neuronal and dopaminergic *Mef2* knockdown markedly increased night activity and reduced sleep, especially under constant darkness, with shorter, more frequent bouts and delayed sleep onset. Conversely, silencing *TRAPPC9* in dopaminergic neurons decreased activity and increased daytime sleep but elevated night activity in constant darkness, while silencing *TRAPPC9* in circadian neurons consistently increased night activity and reduced sleep across light and dark conditions [93].

Regarding environmental influences and factors that could play a role in ADHD development, exposure to Bisphenol A (BPA) during the embryonic period has been shown to induce ADHD-related symptoms [94]. Embryonic BPA exposure reduced the eclosion rate, while the pupae exposed during the larval period to the compound led to smaller body size, morphological abnormalities, and decreased lifespan. Flies exposed to BPA during embryogenesis exhibited elevated aggression and hyperactivity compared to controls, symptoms related to ADHD. These behavioural and physiological alterations are thought to result from reduced dopamine levels and increased production of reactive oxygen species [94].

ADHD treatment varies greatly among patients, particularly in response to methylphenidate (MPH), making it essential to understand this heterogeneity. *D. melanogaster* provides a powerful way to uncover the genetic signatures that drive differences in drug response [95], as well as to pinpoint the specific cell types that respond to MPH or to the non-stimulant drug atomoxetine (ATX), offering valuable insights into treatment mechanisms [96].

Rohde et al. (2019) [95] showed that exposing wild-type flies to MPH increased locomotor activity and altered the expression of genes linked to carbohydrate metabolism. Using the *D. melanogaster* Genetic Reference Panel, they found that behavioural responses to MPH varied widely across genotypes, highlighting strong genetic influences. Network modelling identified 87 predictive gene sets, of which 36 were prioritised, and functional knockdown confirmed contributions of several candidates. Intriguingly, offspring of MPH-exposed parents displayed altered activity without direct exposure, suggesting heritable, likely epigenetic, effects. These findings established that both genotype and intergenerational mechanisms shape drug responsiveness.

At the cellular level, Qu et al. (2024) [96] investigated transcriptional responses to MPH and the non-stimulant atomoxetine (ATX) using single-cell RNA sequencing of adult fly brains. MPH induced broader and stronger effects than ATX, including selective upregulation of the dopamine receptor genes *Dop2R* and *DopEcR* and the dopamine metabolism genes *Syt1*, *Sytα*, *Syt7*, and *Ih*.

Beyond dopamine-related genes, MPH broadly suppressed glutamate and GABA receptor genes as well as synaptic signalling molecules, suggesting wide-reaching synaptic modulation. In contrast, ATX elicited much weaker gene expression changes across these pathways. Together, these studies link population-level genetic variation with cell-type-specific transcriptional mechanisms, demonstrating how inherited and epigenetic factors converge on dopamine pathways and synaptic networks to drive variability in ADHD drug responses [96].

The inattention and hyperactivity present in ADHD patients lead to abnormal social interactions. Parallel phenomena can be modelled in fruit flies, where exposure to the neonicotinoid insecticide imidacloprid disrupted social interaction among flies using the social space assay and increased locomotor activity, moving faster and for longer durations than controls [97]. Imidacloprid also significantly reduces dopamine levels and increases oxidative stress in the fly brain [98]. Moreover, imidacloprid exposure was shown to impair dopaminergic function by decreasing tyrosine hydroxylase (TH) activity. This impairment could be counteracted by supplementation with lutein carrier nanoparticles, which restored TH activity and preserved dopamine signalling [99]. These findings highlight how environmental neurotoxins compromise dopamine synthesis and suggest that targeted neuroprotective strategies may help maintain dopaminergic homeostasis under chemical stress.

ADHD is challenging to model due to its clinical heterogeneity, with symptoms and treatment responses varying widely among individuals. *D. melanogaster* provides a powerful tool for modelling ADHD-related mechanisms. For instance, silencing different genes related to dopamine, including *DAT*, *Latrophilin*, *Nf1*, *Mef2,* or *TRAPPC9* in neuronal populations, reproduces core human ADHD phenotypes, including hyperactivity, reduced sleep, and altered circadian rhythms. Notably, some of these manipulations respond to clinically validated ADHD drugs like MPH, highlighting their translational relevance for elucidating disease mechanisms. Moreover, environmental impacts on dopamine synthesis, such as the neurotoxic effects of insecticides and their rescue by antioxidants, demonstrate the value of flies for studying gene–environment interactions and neuroprotection in ADHD pathophysiology.

### 3.3. Parkinson’s Disease

Parkinson’s Disease (PD) is the second most common neurodegenerative disorder after Alzheimer’s disease, affecting an estimated 6.1 million people worldwide in 2016. It is characterised by uncontrollable motor symptoms, including tremors, rigidity, bradykinesia, and postural abnormalities. which primarily arise from the progressive loss of dopaminergic neurons in the substantia nigra pars compacta (SNpc). While most cases are sporadic, familial forms have been associated with mutations in several genes, including Leucine Rich Repeat Kinase 2 (*LRRK2*) and α-synuclein (*α-Syn*) [100].

*D. melanogaster* has served as a critical model system for studying PD, providing insights into both the genetic and environmental factors contributing to the disease, as summarised in Table 2. Pan-neural expression of human *α-Syn* exclusively in adult flies results in aggregations of ubiquitin conjugated proteins. This overexpression produces PD-like features, including loss of DA neurons, reduced locomotor activity, and shortened lifespan, confirming that DA neuron degeneration can be triggered after development [21]. Directly targeting the expression of *α-Syn* in DA neurons also revealed a key role of ferroptosis in dopaminergic neuron degeneration. Iron supplementation with ferric ammonium citrate (FAC) and glutathione depletion via erastin significantly worsened the phenotype [101]. Conversely, treatments that counteracted ferroptosis offered partial protection. The iron chelator deferiprone (DEF) reduced iron-driven toxicity, while N-acetylcysteine (NAC), a glutathione precursor, restored antioxidant capacity, improving dopaminergic neuron survival, motor behaviour, and lifespan [101]. Additional studies have demonstrated that functional and molecular interactions between α-Syn and parkin contribute to DA neuron degeneration. Simultaneous *α-Syn* overexpression and *parkin* downregulation resulted in a significant loss of dopaminergic neurons in the PPL1 and PPM1/2 clusters. These effects were associated with mitochondrial fragmentation in those clusters, while other dopaminergic ones remained unaffected, highlighting the selective vulnerability within the dopaminergic system. Moreover, *α-Syn* overexpression alone was sufficient to reduce *parkin* transcript levels, suggesting transcriptional regulation as a contributing mechanism [102].

Beyond neuronal factors, glial function is also a determinant of dopaminergic neuron loss. Disruption of the autophagy protein Atg9 in glial cells impaired autophagic flux and led to progressive, age-dependent degeneration of dopaminergic neurons, accompanied by locomotor deficits and glial activation [103].

These α-Syn *D. melanogaster* models have also been employed to investigate molecular changes and metabolic reprogramming associated with PD and ageing. Multi-omic analysis revealed that neuronal *α-Syn* expression induces reprogramming distinct from age-dependent changes. Flies overexpressing *α-Syn* exhibit suppression of glycolysis, insulin signalling, oxidative phosphorylation, and complex I biogenesis, impairing energy metabolism [104]. Together, these findings support a model in which both age-related and α-Syn-driven metabolic changes collaborate to induce PD. Furthermore, the neurotoxic effects of α-Syn involve alterations to the nonsense-mediated decay (NMD) pathway, which is essential for neuronal health [104]. These findings underscore the unique metabolic changes in PD neurons that differ from those associated with normal ageing.

Besides the accumulation of α-Syn, mutations in the *LRKK2* gene are the most common causes of both familial and sporadic PD. These mutations enhance LRRK2 kinase activity, resulting in neurite degeneration and neuronal cell death. In adult fruit flies, overexpression of mutated LRRK2 in DA neurons produced age-dependent neuron loss. This effect is mediated by elevated translation of Furin1 (Fur1) in DA neurons, which enhances Gbb-mediated activation of BMP signalling in glia, leading to DA neurodegeneration. Reducing Fur1 levels rescued DA neuron survival, climbing behaviour, and fly lifespan [105]. Interestingly, knockdown of Gbb or BMP pathway components in glia, but not neurons, significantly prevented DA neuron loss. Besides that, exposure to the herbicide paraquat similarly increased Fur1 translation and activated BMP signalling in glia, mirroring effects seen with *LRRK2* mutation, suggesting that genetic and environmental factors could converge to generate the disease [105].

Further research revealed that the expression of mutant forms of *LRRK2* in glia also caused age-dependent degeneration of DA neurons in PPL1 clusters, accompanied by locomotor deficits. This deficient glia induced neuroinflammatory responses, but pharmacological intervention with levetiracetam attenuated these effects, rescuing dopaminergic survival, restoring motor performance, and reducing inflammatory markers [106].

PD is generally regarded as an age-related disorder. However, mutations in *PINK1* are associated with early-onset cases, in which symptoms typically appear before the age of 50. Knockout flies of *Pink1 (Pink1 B9)* exhibit electrophysiological defects in dopamine PPM3 cluster neurons, leading to increased spontaneous activity and altered properties of action potentials. This disrupted PPM3 neuronal function generates motor impairments in *Pink1*-deficient flies [107]. Additionally, brain dopamine levels are reduced, and the lifespan of *Pink1 B9 flies* is shorter than WT controls [108] (CITA Xiao-Lin Bai 2023).

In this context, the neuroprotective effects of *Lycium barbarum* fruit extract (LBFE) and ginseng total protein (GTP) against *Pink1*-related PD symptoms have been investigated. Both LBFE- and GTP-treated *Pink1 B9* flies had an improved lifespan compared to untreated ones, as well as improved locomotor ability. GTP-supplemented diet reduced dopamine neuron loss, whereas LBFA increased dopamine brain levels [108,109].

PD can also be induced in flies using the pesticide rotenone [110,111]. Exposure to the pesticide induces changes in locomotor behaviour in a time-dependent manner without generating DA neuron loss [110]. Rotenone-exposed flies exhibit a reduction in tyrosine hydroxylase (TH) synthesis in DA neurons, the rate-limiting enzyme for dopamine production, which generates decreased DA levels in the brain, accompanied by a reduction in 3,4-dihydroxyphenylacetic acid (DOPAC) and an increase in Homovanillic acid (HVA), which are DA metabolites, indicating increased oxidative turnover of dopamine [111]. Interestingly, rotenone also induces transcriptomic changes in DA neurons in an early phase of PD, associated with cell death and neuronal function. Among the molecular changes, key signalling pathways such as MAPK/EGFR and TGF-β were upregulated, while the Wnt signalling pathway was notably suppressed [110]. Notably, Wnt signalling rescued the rotenone-induced locomotor deficits, suggesting that Wnt pathway impairment is a driving mechanism in DA neuron pathology [110].

Fruit flies have been instrumental in elucidating the molecular and cellular mechanisms underlying PD, particularly the progressive degeneration of DA neurons that underlie motor dysfunction. These models recapitulate key human PD phenotypes, such as loss of DA neurons, reduced locomotor activity, shortened lifespan, and neuroinflammatory responses, which emphasise conserved pathogenic pathways with mammals. Some examples of these conserved pathogenic pathways include the generation of mouse models of PD generated by mutations in the *LRRK2* gene [112] or exposing mice to rotenone [113], similar to *D. melanogaster* models. Different treatments tested in flies have also been tested and used in PD patients or mouse models, including NAC treatment [114] and levetiracetam [115]. Overall, *D. melanogaster* represents a powerful and versatile tool for deciphering the molecular mechanisms underlying PD and for evaluating the efficacy of therapeutic compounds and mechanistic interventions in a conserved biological context.

## 4. Conclusions

Recent advances in modelling diseases related to peripheral and dopamine systems highlight *D. melanogaster* as a valuable resource for exploring mechanisms behind neurological disorders. Its significant genetic and functional similarities with humans enhance its role in translational research and for cost-effective preclinical applications.

These models not only facilitate genetic dissection but also enable in vivo investigation of disease-associated biomolecules through genetic and pharmacological approaches. Such approaches can deepen our understanding of molecular pathways and accelerate the discovery of new therapeutic targets. Moreover, the automation of behavioural tests in flies, powered by artificial intelligence and machine learning, can significantly accelerate gene and drug discovery, opening opportunities to treat and reduce the burden of these neurological disorders.

Additionally, the recently completed *D. melanogaster* brain connectome offers exciting opportunities for mapping neural substrates and synaptic connections linked to disease phenotypes. By integrating connectomics information with molecular analyses, we can effectively link genotype, neural circuitry, and behaviour in the context of diseases.

In conclusion, *D. melanogaster* provides an integral model for unravelling the molecular and cellular bases of neurological disorders, with ongoing advancements promising to shape future discoveries and therapeutic interventions significantly.

## Figures and Tables

**Figure 1 biomolecules-15-01677-f001:**
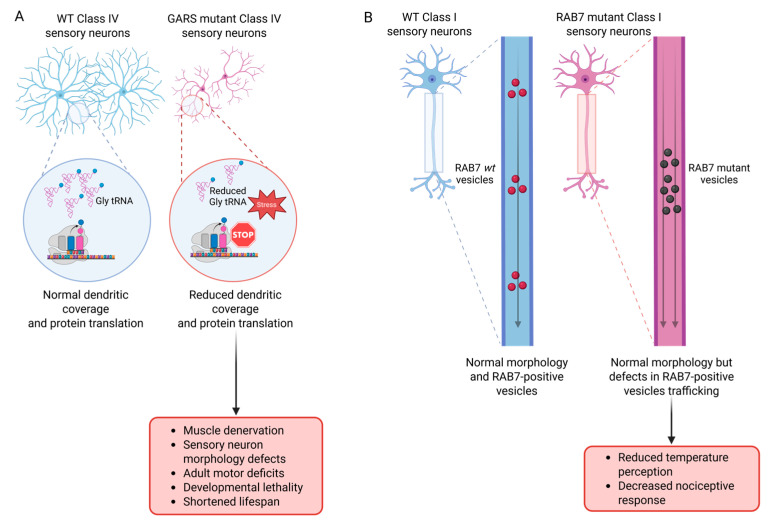
Pathogenic mechanisms of GARS and Rab7 mutations in CMT models. (**A**) In wild-type C4da, sufficient levels of Gly tRNA allow for normal dendritic coverage and protein translation. In contrast, GARS mutant neurons show reduced Gly tRNA availability, leading to ribosome stopping at glycine codons, activation of the stress response, reduced protein synthesis, and decreased dendritic coverage, generating CMT-related phenotypes (bottom panel). (**B**) Wild-type Rab7 promotes normal axonal trafficking of Rab7-positive vesicles without affecting neuronal morphology. Mutant Rab7 causes abnormal accumulation and defective trafficking of Rab7-positive vesicles, resulting in sensory impairments despite preserved neuronal morphology that gives rise to sensory defects (bottom panel). Created with BioRender, https://BioRender.com/ (accessed on 7 October 2025).

**Figure 2 biomolecules-15-01677-f002:**
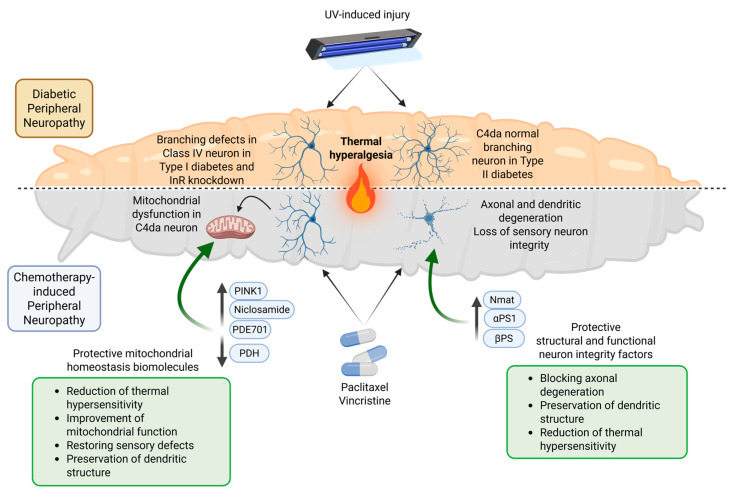
*D. melanogaster* models of diabetic and chemotherapy-induced peripheral neuropathy. (**Top panel**) Type I and II diabetes, or *InR* knockdown, lead to persistent thermal hyperalgesia after UV-induced injury. In type I diabetes and *InR* knockdown, C4da neurons show reduced dendritic branching, whereas type II diabetes preserves morphology. (**Bottom panel**) Chemotherapy with paclitaxel or vincristine causes axonal and dendritic degeneration, mitochondrial dysfunction, and nociceptive hypersensitivity. Protective mechanisms include mitochondrial regulators that restore mitochondrial homeostasis and integrity factors that preserve dendritic architecture and neuronal function. These interventions reduce thermal hypersensitivity, block axonal degeneration, and restore sensory, among other defects. The models highlight conserved pathways and potential therapeutic targets for diabetic and chemotherapy-induced neuropathies. Created with BioRender, https://BioRender.com/ (accessed on 7 October 2025).

**Figure 3 biomolecules-15-01677-f003:**
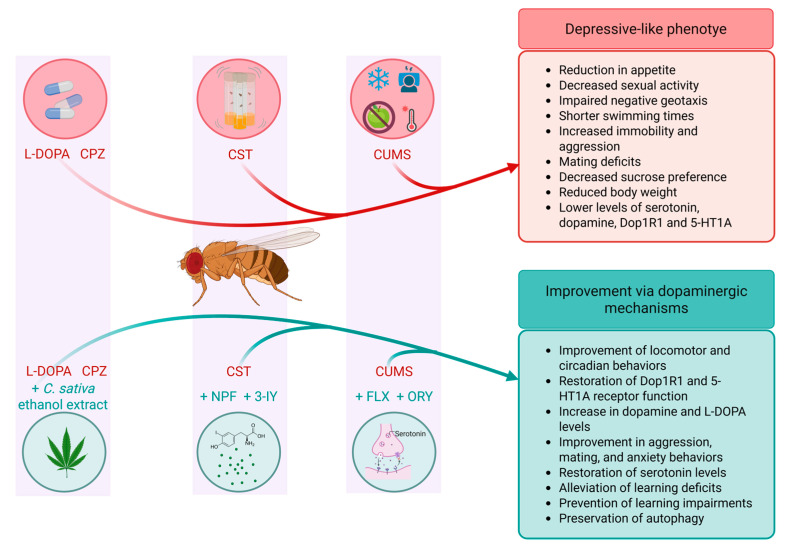
*D. melanogaster* models of depression-like disorders. Depression-like phenotypes in *D. melanogaster* can be induced through pharmacological treatments (e.g., L-DOPA, CPZ), chronic unpredictable mild stress (CUMS), or chronic stress treatment (CST). These manipulations result in behavioural and physiological alterations, (**top panel, right**). Antidepressant interventions shown to ameliorate these phenotypes include *Cannabis sativa* ethanol extract, 3-IY, NPF, FLX, and ORY (**bottom panel, right**). Created with BioRender, https://BioRender.com/ (accessed on 7 October 2025).

**Table 1 biomolecules-15-01677-t001:** Peripheral neuropathy models using the larval peripheral nervous system of *D. melanogaster*.

Disease Model	Genetic/Chemical Manipulation	Observed Phenotypes	Validated Interventions
CMT—*GARS* mutant	Expression of mutant *GARS* in motor and C4da neurons	Disrupted neuromuscular junction integrity; progressive muscle denervation; impaired neuronal function via translational dysregulation; reduced dendritic coverage; decreased protein translation; developmental lethality; shortened lifespan	Overexpression of Glycine tRNA partially restores translation, reduces denervation, improves sensory morphology, motor function, and survival
CMT—*RAB7A* mutant	Expression of human mutant *Rab7* in sensory neurons	Reduced temperature perception; decreased nociceptive response; no dendritic alteration; accumulation of Rab7-positive vesicles in axons; reduced vesicle stationary time	None reported in flies
DPN—InR mutant	InR mutants	Persistent thermal hyperalgesia	None reported in flies
DPN—Diabetes type 1	Silencing the insulin-producing cells	Persistent thermal hyperalgesia; reduced C4da dendritic length and branch number	None reported in flies
DPN—Diabetes type 2	High sugar diet	Persistent thermal hyperalgesia; no major dendritic changes	None reported in flies
DPN—Sensory neuron-specific	*InR* knockdown in md neurons	Persistent thermal hyperalgesia; reduced baseline dendritic length; elevated calcium responses in C4da neurons	Restoring insulin-like signalling in md neurons rescues persistent nociceptive hypersensitivity
CIPN—Paclitaxel	Feeding larvae paclitaxel	Thermal hypersensitivity; C4da dendritic defects; axonal loss; disrupted integrin-mediated adhesion and trafficking, causing dendritic branch crossing; mitochondrial dysfunction.	*Pink1* overexpression in C4da neurons restores mitochondrial homeostasis and alleviates thermal hypersensitivity. Niclosamide and PDE701 similarly improve mitochondrial function, with PDE701 additionally reducing thermal hyperalgesia. Nmnat overexpression prevents axonal degeneration and mitigates nociceptive hypersensitivity. Enhanced expression of αPS1 and βPS integrins preserves dendritic integrity and protects against thermal sensitivity.
CIPN—Vincristine	Feeding larvae vincristine	Thermal hypersensitivity; structural alterations in C4da neurons; mitochondrial and sensory dysfunction	PDH knockdown rescues mitochondrial and sensory defects
ALS—TBPH	Upregulation of *TBPH* Downregulation of *TBPH* G287S *TDP-43* mutant	Excessive dendritic branching in C4da neurons (overexpression); reduced dendritic branching in C4da neurons (downregulation); altered nuclear–cytoplasmic localization (G287S mutant impairs calcium-dependent transport)	None reported in flies
ALS—Calcium-dependent regulators	Silencing Calmodulin, Protein kinase C, or CalpainA in sensory neurons	TBPH cytoplasmic retention in pupal sensory neurons	Overexpression of Importin α3 and β1 enhances nuclear translocation
ALS—*C9orf72*	Expression of UAS-(GGGGCC)48 RNA in C4da neurons	Late-stage severe dendritic loss; defective local RNA translation	Silencing *dFMR1* or *Orb2* rescues dendritic branching defects. Overexpression of *dFMR1* or *CrebA* increases the branching abnormalities
ALS—DPRs (PR36, GR36)	Expression of *PR36* or *GR36* in C4da neurons	Loss of dendritic branch points; reduced GOPs; impaired plasma membrane supply; PR36 causes *CrebA* downregulation	*CrebA* overexpression restores GOPs and plasma membrane supply in neurons that express *PR36* but fails to recover dendritic morphology or prevent degeneration
ALS—Sod1 mutant	Expression of mutant *Sod1*	Early stage: sensory feedback defects preceded motor degeneration; late stage: motor neuron degeneration; motor impairment	BMP signalling activation in proprioceptive neurons rescues early and partially late phenotypes

**Table 2 biomolecules-15-01677-t002:** CNS neuropathy models using the dopaminergic system of adult *D. melanogaster*.

Disease Model	Genetic/Chemical Manipulation	Observed Phenotypes	Validated Interventions
Depression-like—L-DOPA	Feeding larvae L-DOPA	Reduction in appetite; decreased sexual activity; negative geotaxis deficits; *CG4269* upregulation in males; downregulation of *CG6821* in females	None reported in flies
Depression-like—CPZ	Feeding larvae CPZ	Reduction in appetite; decreased sexual activity; *CG4269* upregulation and *CG6821* downregulation in males; reduced Dop1R1 and 5-HT1A; lower dopamine and L-DOPA levels	*C. sativa* HE improves locomotor and circadian behaviours, restores Dop1R1/5-HT1A, and increases dopamine/L-DOPA
Depression-like—CUMS	Chronic unpredictable stressors (10 days of multiple stressors: heat, cold, sleep deprivation, starvation)	Shorter swimming times; increased immobility; heightened aggression; mating deficits; decreased sucrose preference; reduced body weight; more time in dark box; lower serotonin and dopamine	FLX and ORY improve aggression, mating, anxiety, restore serotonin and partially dopamine levels
Depression-like—CST/CSLD	Repeated mechanical stress	Impaired olfactory learning and memory; behavioural impairments; maladaptive dopaminergic signalling; disrupted autophagy flux in neurons	3-IY alleviates learning deficits. Blocking DA neuron transmission prevents learning deficits. NPF signalling confers resilience, preserving autophagy
ADHD—*DAT*, *LPHN3*, *NF1* knockdown	Pan-neuronal knockdown of *DAT*, *LPHN3*, *Nf1*	Elevated activity; reduced sleep during darkness	MPH reversed behavioural abnormalities
ADHD—*Mef2* and *TRAPPC9* knockdown	Knockdown of *Mef2* and *TRAPPC9* in pan-neuronal, dopaminergic, or circadian populations	Pan-neuronal and dopaminergic *MEF2* knockdown: Increased night activity; reduced sleep under constant darkness; delayed sleep onset. *TRAPPC9* silencing in dopaminergic neurons: decreased activity; increased daytime sleep with elevated activity in darkness. *TRAPPC9* silencing in circadian neurons: increased night activity and reduced sleep	None reported in flies
ADHD—Bisphenol A	BPA feeding parental flies	Reduced eclosion rate; smaller body size, morphological abnormalities, and decreased lifespan in pupae; elevated aggression and hyperactivity; reduced dopamine levels; increased production of reactive oxygen species.	None reported in flies
ADHD—MPH	Exposing flies to MPH	Increased locomotor activity; altered carbohydrate metabolism; selective upregulation of dopamine receptor genes and dopamine metabolism genes; suppression of glutamate and GABA receptor genes; suppression of synaptic signalling molecules	None reported in flies
ADHD—imidacloprid	Exposing flies to imidacloprid	Disruption of social interaction; increased locomotor activity; reduction in dopamine levels; oxidative stress increase; impairment of dopaminergic function by decreasing TH activity	Lutein carrier nanoparticles restore DA synthesis and TH activity
PD—α-Syn	Pan-neural expression of human *α-Syn*	Aggregation of ubiquitin-conjugated proteins; loss of DA neurons; reduced locomotor activity; shortened lifespan; reduction in parkin transcript levels; suppression of glycolysis, insulin signalling, oxidative phosphorylation, and complex I biogenesis; impairing energy metabolism; alterations in NMD pathway	DEF and NAC restore antioxidant capacity, improving dopaminergic neuron survival, motor behaviour, and lifespan
PD—α-Syn and parkin	*α-Syn* overexpression and parkin downregulation	Significant loss of dopaminergic neurons in the PPL1 and PPM1/2 clusters; mitochondrial fragmentation	None reported in flies
PD—*Pink1 B9*	*Pink1 B9*	Dopamine PPM3 electrophysiological defects; motor impairments; reduced brain dopamine levels; reduced lifespan.	LBFE and GTP improve lifespan, locomotor ability, reduce dopamine neuron loss, and increase dopamine brain levels.
PD—*Atg9*	Silencing *Atg9* in glial cells	Impaired autophagic flux; progressive, age-dependent dopaminergic neuron degeneration; locomotor deficits; glial activation	None reported in flies
PD—Mutated *LRRK2*	Overexpression of mutated *LRRK2* in DA neurons	Age-dependent neuron loss; elevated Fur1 in DA neurons; activation of BMP signalling in glia; DA neurodegeneration	Furin1 reduction rescues DA neuron survival, climbing behaviour, and lifespan; knockdown of Gbb or BMP pathway components in glia prevents DA neuron loss.
PD—paraquat	Exposure to paraquat	Age-dependent neuron loss; elevated Fur1 in DA neurons; activation of BMP signalling in glia; DA neurodegeneration	Reducing Fur1 or BMP pathway prevents DA neuron loss
PD—Mutated *LRRK2*	Overexpression of mutated *LRRK2* in glia	Age-dependent DA neuron loss in PPL1 clusters; locomotor deficits; neuroinflammatory responses	Levetiracetam exposure restores dopaminergic survival, locomotor performance, and reduces inflammatory markers.
PD—rotenone	Exposure to rotenone	Changes in locomotor behaviour without DA neuron loss; reduction in TH synthesis in DA neurons; decreased DA levels in the brain; reduction in DOPAC and increase in HVA; increased oxidative dopamine turnover; upregulation of MAPK/EGFR and TGF-β pathways; Wnt pathway suppression.	Wnt pathway activation rescues motor phenotype

## Data Availability

No new data were created or analysed in this study. Data sharing is not applicable to this article.

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
