# Peer review of "Modelling Neural Disorders with the D. melanogaster Larval Peripheral and Adult Dopaminergic Systems"

_biomolecules, 2025, doi:10.3390/biom15121677_

Round 1
Reviewer 1 Report
Comments and Suggestions for Authors
Thank you for submitting this interesting and comprehensive review on the use of Drosophila melanogaster models to study neurodegenerative and neuropsychiatric diseases. The manuscript addresses a timely topic and effectively highlights how larval and adult models complement each other in uncovering disease mechanisms and screening potential therapeutics.
However, to improve the scientific rigor, readability, and translational relevance of the paper, I recommend the following revisions:
-
Indicate whether this is a narrative or scoping review and briefly describe how studies were selected (databases, keywords, time frame).
-
The Parkinson’s disease section is detailed, but depression and ADHD sections are comparatively brief.
-
Include tables summarizing disease models, genetic or chemical manipulations, observed phenotypes, and validated interventions.
-
Conclude each disease section with short “translational insights” summarizing how findings in flies map to mammalian or clinical systems.
-
Add measurable endpoints or behavioral outcomes to figures for better interpretability.
-
Standardize gene and protein nomenclature (e.g., LRRK2, α-synuclein), fix typographical inconsistencies, and define all abbreviations at first mention
Reviewer 2 Report
Comments and Suggestions for Authors
The authors provide a clear and balanced overview of Drosophila models, highlighting the strengths of larval peripheral and adult dopaminergic systems for studying human-relevant neurological diseases and their potential to further translational neuroscience research.
The figures are well-designed and informative, and I appreciate the clarity of the data presentation. However, increasing the font size would further enhance readability, particularly in the printed version.
Figure 3: The legend refers to panels A and B, but these panels are not present in the figure. This discrepancy should be addressed.
PINK1 is mentioned in the paragraph discussing CIPN models, but is not addressed in the Parkinson’s disease section, despite several studies utilising the Drosophila model.
Line 77: The value “17.69/100000” should be reformatted for improved readability, for example, as “17.69 per 100,000.”
While the abbreviations used are standard, it would be helpful to spell them out at least once for clarity. For example:
Line 297: DEG (Differentially Expressed Genes)
Line 392: GWAS (Genome-Wide Association Study)
Line 461: The sentence “Additional studies have demonstrated that interactions between α-Syn and parkin contribute to DA neuron degeneration” could be misinterpreted as suggesting a direct physical interaction between the proteins. Consider clarifying the nature of these interactions (e.g., functional, genetic, or pathway-level interactions) to avoid ambiguity.
Reviewer 3 Report
Comments and Suggestions for Authors
See attached file.

The English writing is good, but the few editing corrections listed in my report are needed.
Round 2
Reviewer 1 Report
Comments and Suggestions for Authors
Well done